# Role of Pre-Harvest Sorbitol–Calcium Treatments in Controlling Berry Drop in Bagged Table Grapes of the "Doña María" Variety

Alberto Guirao, Juan Miguel Valverde *, Huertas María Díaz-Mula, Daniel Valero, María Serrano and Domingo Martínez-Romero *

Institute for Agro-Food and Agro-Environmental Research and Innovation (CIAGRO), University Miguel Hernández (UMH), Ctra. Beniel Km. 3.2, 03312 Orihuela, Alicante, Spain; a.guirao@umh.es (A.G.); h.diaz@umh.es (H.M.D.-M.); daniel.valero@umh.es (D.V.); m.serrano@umh.es (M.S.)
* Correspondence: jm.valverde@umh.es (J.M.V.); dmromero@umh.es (D.M.-R.)

**Abstract:** Cv 'Doña María' table grape is a high-quality table grape variety included in the Protected Designation of Origin (PDO) of the European Union "The bagged grape of the Vinalopó". The PDO stipulates that grape clusters must be protected with paper bags from inclement weather and insects, which helps enhance the final grape quality. However, 'Doña María' is a variety prone to high shattering in the late stages of ripening on the vine and during postharvest. Inorganic calcium treatments are one of the most commonly used tools to reduce this disorder, but the translocation of this mineral from veraison onward has been questioned. In this study, five applications were performed, from veraison to harvest, using sorbitol-chelated calcium (0.7% + 1.0%), Ca(NO$_3$)$_2$ (Ca) at 0.7% and sorbitol at 1%. It was observed that bagged grapes (not wetted with the solutions) only increased the concentration of total and bound calcium when treated with sorbitol–Ca. This resulted in reduced berry drop during cultivation and postharvest and improved fruit firmness. Additionally, it reduced grape metabolism indicators such as respiration rate, weight loss, malic acid degradation, prevented abscisic acid (ABA) and malondialdehyde (MDA) accumulation, and favored the accumulation of secondary metabolites such as total polyphenols, increased antioxidant activity, and sugar content. The application of sorbitol-chelated calcium is an effective and safe tool that enhances fruit quality and prevents losses due to shattering during postharvest.

**Keywords:** *Vitis vinífera*; polyol; postharvest; shattering; ABA; MDA; organic acids; polyphenols; sugars





## 1. Introduction

Table grape (*Vitis vinifera* L.) is a widely cultivated and consumed fruit worldwide due to its sweet and juicy taste and nutritional value. However, during cultivation and postharvest life, table grapes face various challenges. Among these challenges, reducing grape losses at the time of harvest due to lack of quality (size, color, etc.) or spoilage or berry drop are significant concerns. Additionally, in postharvest, various issues arise, such as berry detachment from the stem, softening, loss of stem color, and the onset of rot. These problems discard a high percentage of fruits before consumption, leading to significant economic losses for producers and along the marketing chain [1–3]. Therefore, it is necessary to implement strategies (healthy, sustainable, and efficient) to improve production, enhance and maintain quality, and reduce food waste.

The table grape variety 'Dona María' is a white variety with large cylindrical clusters. The berries are quite large, elliptical, and turn yellowish-green upon ripening, originating from a cross between the varieties 'Moscatel de Setubal' and 'Rosaky' [4]. This grape variety reaches maturity from late August to the second week of October in the southeast of Spain. Its main characteristics include a resistant skin covered with a thick layer of bloom and a firm, sweet, and juicy pulp. However, the worst defect of the 'Dona Maria' table grape is

its susceptibility to berry drop during ripening, which is the main challenge for storage and marketing [5].

The cultivation of the 'Dona María' table grape variety is included in the list of varieties allowed by the Protected Designation of Origin (PDO) "Uva de mesa embolsada Vinalopó" [6]. At the time of harvest, a minimum of 12.5 °Brix in total soluble solids (SST) is required. A distinctive feature of this PDO "Uva de mesa embolsada Vinalopó" is that the clusters remain individually protected by a paper bag from veraison until harvest. The bag protects the cluster from insect attacks, birds, and minor weather incidents, improves the coloration of grape berries, and delays their ripening [6]. The cluster is placed inside the bag, open at both ends, and secured to the peduncle of the cluster, leaving it open at the bottom. At the time of harvest, the grapes, protected by the PDO, will have remained shielded by the bag for a minimum of sixty days.

The dropping of berries from the stem (shattering) is a concerning phenomenon that occurs after harvest and during the handling and transportation of table grapes. This problem can arise from several factors: (1) berries detaching from the rachis because of the delicate structure of the stem tissue; (2) wet drop, where berries separate from the stems but stay attached to the pedicel due to the short, thin brush; and (3) dry drop or abscission, resulting from the development of an abscission zone (AZ) in the berry [7] (Deng et al., 2007). Berry drop not only reduces crop yield but also affects the appearance and quality of marketable grapes [8]. It has been demonstrated that shattering is correlated with the degradation of pectins and celluloses in the abscission zone and the increase in the activity of hydrolytic enzymes [7]. Tissues with a better structure, with calcium bound to the carboxylic groups of pectins, could allow greater berry adherence and reduce their drop.

Berry drop is influenced by genetic factors, as well as by the degree of grape ripening [1]. In addition, the use of different cultivation techniques and the use of growth regulators like abscisic acid (ABA) [9] or 2-chloroethylphosphonic acid (Ethephon) [10] plays a role in berry shattering. Climatic conditions during cultivation, such as the increase in daily average temperatures or extreme temperatures (40 °C) during ripening on the vine [11], as well as postharvest storage conditions favoring cluster weight losses [12] or improper handling of clusters during marketing [13], should also be considered.

Various pre-harvest treatments have been successfully used to prevent berry shattering, such as 6-benzylaminopurine (6-BA), 2,3,5-triiodobenzoic acid (TIBA), gibberellic acid (GA$_3$), indole acetic acid (IAA), naphthalene acetic acid (NAA), potassium permanganate (KMnO$_4$), ascorbic acid (AA), and cyanocobalamin (B12) [9,12]. However, the use of some of these successfully tested products is limited or legally banned in some countries. Faced with these challenges in table grape, effective and healthful strategies are required to improve their quality and shelf life. In this regard, the use of calcium treatments has emerged as a possible solution. Calcium is an essential and principal nutrient for plants, required as a divalent cation (Ca$^{+2}$) in a wide variety of functions, including structural function in the cell wall and membranes and as a cytoplasmic secondary messenger related to environmental or developmental stimuli for physiological responses [14].

The cell wall, composed of cellulose, hemicellulose, pectins, and glycoproteins, has mechanical and protective functions in plant cells. Specifically, during fruit ripening, hydrolytic enzymes alter the structure of pectins, making the fruit more susceptible to softening, physiological disorders, and pathogen attacks. Calcium mitigates postharvest disorders by fortifying cell walls and preserving membrane integrity and selective permeability [15]. Applying calcium helps maintain cell turgor and tissue firmness while delaying the breakdown of membrane lipids, thereby extending the shelf life of fresh fruits [16–18].

In grape berries, high calcium levels resulting from pre- and post-veraison Ca treatments have been shown to delay senescence and enhance resistance to *B. cinerea* [19]. For example, applying calcium chloride to table grape clusters after veraison has been effective in reducing *B. cinerea* rot, yielding positive outcomes postharvest and maintaining grape protection for up to six weeks in cold storage [20]. Sabir and Sabir [21] demonstrated the essential role of calcium in preserving the quality and shelf life of table grapes during

postharvest storage. This effect is attributed to the ability of calcium to slow weight loss, reduce decay, maintain rachis chlorophyll levels, and preserve visual quality during extended periods of cold storage.

Numerous studies have demonstrated that calcium continues to accumulate in grapes during their development [22,23], whereas other research suggests that this accumulation ceases after veraison [24]. This cessation is attributed to the loss of functionality in the xylem vessels within the rachis [25]. Furthermore, calcium uptake through the berry skin significantly diminishes due to the loss of stomatal functionality [26]. Calcium dissolved in the xylem fluid initially arrives to adult leaves, moved by transpiration stream, and thereafter, poor translocation occurs from leaves to young growing leaves or fruits [27,28]. Calcium content in berries depends on the calcium fertilizer applied and on the berry development stage at the time of treatment [19,29].

Calcium compounds can be classified according to their solubility and physiological activity, including transport possibilities in the plant. For instance, soluble calcium exists as nitrates, chlorides, and organic acids; exchangeable calcium may be linked to carboxylic groups of pectins or bound to proteins [27]; calcium that lacks physiological activity is present in the form of calcium oxalate, phosphates, and carbonates [30].

Polyols, including cis-diol groups, such as sorbitol and mannitol, can form stable compounds with other metabolites, facilitating their transportation through the phloem. This has been reported for minerals like Zn, Ca, B, and salicylic acid [31–37]. Most of these aforementioned metabolites have low mobility in the phloem. Polyols combined with metabolites can be foliar-absorbed and directed by a concentration gradient to reach phloem cells, and their translocation and accumulation in different target organs are increased [34,38]. For instance, a higher concentration of Zn has been observed in apples treated with sorbitol–Zn compared to apples treated with Zn alone [31], and the sorbitol–Ca complex led to increased Ca accumulation in fruits, meristems, leaves, and roots of peanut plants [33]. Thus, the application of calcium in combination with polyols may be a new technology facilitating calcium mobility through the plant [34,38] and preventing physiological disorders and pathogen attacks due to calcium deficiency.

According to the previous comments, it was hypothesized that the use of sorbitol combined with calcium (Ca) applied as pre-harvest treatments to the vines during berry development could improve production (reduce losses during harvest) and maintain postharvest quality of table grapes by reducing the occurrence of rot, berry detachment from clusters, berry cracking, softening, and controlling stem discoloration.

Therefore, the objective of this study is to evaluate whether calcium either applied individually as inorganic calcium (Ca(NO₃)₂) (Ca) or in combination with sorbitol (sorbitol–Ca) can translocate to grapes and prevent their detachment from clusters during cultivation and postharvest storage. This information is easier to determine in bagged grapes grown under the conditions required by the PDO. Bagging prevents treatment contact with the berries; thus, the effect of calcium treatments would be limited solely to the calcium transported from the leaves and accumulated in the berries. In this regard, this is the first time that pre-harvest treatments with calcium chelated in sorbitol are applied to table grapes, and their role in reducing grape detachment from clusters during cultivation and postharvest is evaluated. Additionally, the role of this treatment on grape ripening and quality loss during storage is assessed.

## 2. Materials and Methods

### 2.1. Plant Material and Experimental Design

Grape (*Vitis vinifera* L.) cv 'Doña María' was cultivated in high trellises under the standards of the PDO Table Grape "Vinalopó" [6]. The vines used in the experiment were seven years old and exhibited vigorous and healthy vegetative growth. The study plot was located in the municipality of Hondón de las Nieves (Alicante, Spain) (UTMX: 686,059.00, UTMY: 4,243,420.00) under Mediterranean climatic conditions with high interannual irregularity in precipitation, leading to periods of drought alternating with episodes of heavy

rains that can cause flooding. The climatic conditions during the experimental period were recorded by the weather station from "Monforte del Cid" [39], which is close (16 km) to the field experiment (Monforte del Cid-Alicante, UTMX: 698,193.00, UTMY: 4,252,510.00, elevation 259 m) (Figure 1). The planting density was 1200 vines ha$^{-1}$. The trellising system consisted of wires forming grids of 1 m$^2$ above the vine, to which the shoots intertwine and the grapes hanged from this structure, allowing a large surface leaf area to be exposed to sunlight. The grape clusters were bagged before veraison (24 July 2023) by using a cellulose paper bag, as prescribed by the PDO procedure.

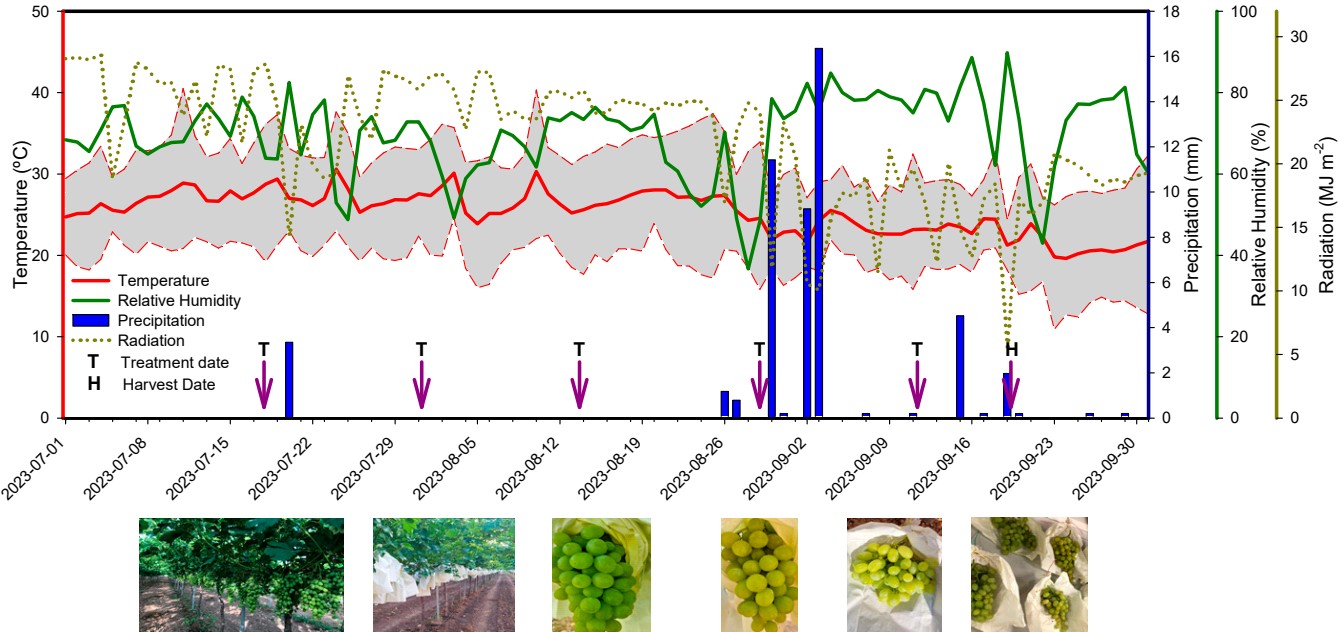

**Figure 1.** Climatological data collected by the IVIA weather station located in Monforte del Cid-Alicante (Spain) from 1 June to 30 September 2024. Dates of treatment application and cluster harvesting.

Treatments were carried out by using freshly prepared solutions of 1% sorbitol (sorbitol) (Barcelonesa de Drogas and Productos Químicos SAU, Barcelona, Spain), 0.7% calcium nitrate (Ca) (Soluteck, Fertiberia SA, Murcia, Spain), sorbitol–calcium nitrate complex at 1% and 0.7%, respectively (sorbitol–Ca), and control vines were treated with tap water (Control). In addition, a non-ionic surfactant, polyglycol alkyl 20% $w/v$ (Elogium, Sipcam Iberia SL, Valencia, Spain), was added as a co-adjuvant in all treated solutions and in control. The first application was performed before veraison and bagging of the clusters on 17 July, and thereafter, treatments were repeated every 15 days until 11 September, 8 days before harvesting (five applications in total, Figure 1). Each treatment was carried out in a row of 120 vines with a dose of 1200 L ha$^{-1}$. Each treated and control row was separated by two untreated ones. Additionally, ten vines at the ends of each row were not considered for the experiment. Treatments were applied with foliar spray by using a sprayer tank connected to a tractor. The tractor speed, pump pressure, and nozzle opening were adjusted so that each vine received one liter of solution.

Fifty bagged clusters were harvested at random from the vines of each treatment on September 19th (when commercial maturation stage was achieved in control vines, according to commercial practices), as well as 100 leaves, and transferred to the laboratory in two hours under refrigerated conditions. Additionally, the number of vines with dropped berries on the ground was counted. In the laboratory, the leaves from each treatment were divided into three batches and dehydrated to measure their mineral content. The paper bags covering the clusters were removed. Well-formed clusters, with an average weight of 766.75 ± 46.99 g, regular berry size (23.06 ± 0.11 mm equatorial diameter and 31.18 ± 0.18 mm longitudinal diameter), were selected. Berries with defects were removed

from the clusters. Three batches, consisting of 3 replicates of 3 clusters, were prepared from each treatment, which were packaged in plastic containers and stored at 1 °C and relative humidity of 95%. Three clusters from each treatment were sampled at random, at day 0 and after 15 and 30 days of storage for the following analytical determinations.

### 2.2. Berry Shattering

Berry detachment from the clusters was assessed both at harvest, by enumerating vines with detached grapes on the ground (>10 berries), and postharvest, by quantifying the percentage of detached berries from the clusters after a brief 5 s shake of each cluster held by the peduncle on each sampling day. Postharvest shattering was computed at the conclusion of storage by summing the number of fallen berries divided by the total number of berries in each cluster, multiplied by 100.

### 2.3. Fruit Respiration Rate and Weight Loss

The respiration rate of grape clusters was assessed using a closed static system. Each treatment was given three 5 L containers with airtight lids, each containing three clusters for one hour. Gas samples (1 mL) were extracted from the airspace within each container for $CO_2$ analysis via chromatography at 20 °C. The fruit's respiration rate was measured using a Shimadzu CG-14B gas chromatograph, which featured a thermal conductivity detector and a 2 m by 1/8 inch CHROMOSORB 102 80/100 column, following the procedure described by Valverde et al. [40]. The results were reported in mg $CO_2$ kg$^{-1}$ h$^{-1}$, and the mean $\pm$ SE was calculated for each set of three replicates.

Weight loss was determined by measuring the difference between the weight of each bunch at harvest and its weight after each storage interval, then dividing by the initial harvest weight. The results were expressed as a percentage of the initial weight (%), and the data were presented as the mean $\pm$ standard error (SE) of three replicates.

### 2.4. Quality Grape: Firmness, Total Soluble Solids (TSS), Titratable Acidity (TA), and Maturation Index (MI)

The firmness of grapes was assessed using a TX-XT2i Texture Analyzer (Stable Microsystems, Godalming, UK) connected to a personal computer, following the protocol outlined by Lorente-Mento et al. [41]. A flat probe with a diameter of 10 cm applied the necessary force to deform the equatorial zone of each whole berry (pedicel, flesh, and peel) by 5%. Results were expressed as the ratio of force to distance covered (N mm$^{-1}$), with data representing the mean $\pm$ SE of three replicates. Each replicate comprised 45 berries from 3 clusters.

To determine TSS and TA, a sample of 10 berries was taken from each cluster (30 berries per replicate). Juice extraction was conducted by squeezing berries through a cotton cloth. TSS was assessed in duplicate at 20 °C using a digital refractometer (Atago PR-101, Atago Co. Ltd., Tokyo, Japan) and presented as grams per 100 g of fresh weight. For TA determination, also performed in duplicate for each sample, 1 mL of juice was diluted to 25 mL with distilled water and titrated to pH 8.1 using an automatic titrator (785 DMP Titrino, Metrohm) with 0.1 N NaOH. Results were expressed as g 100 mL$^{-1}$ of tartaric acid equivalents on fresh weight (FW). The maturation index (MI) was derived as the ratio of TSS to TA. Data were reported as the mean $\pm$ SE of three replicates.

### 2.5. Organic Acids and Sugar Content

The grape juice was subjected to centrifugation at 10,000$\times$ *g* for 10 min. Following this, the supernatant underwent filtration through a 0.45 μm Millipore filter before being introduced into an HPLC system (Hewlett-Packard HPLC series 1100, Agilent, Madrid, Spain) for the quantification of individual sugars and organic acids. The elution system utilized 0.1% phosphoric acid running isocratically at a flow rate of 0.5 mL min$^{-1}$ through a Supelco column (Supelcogel Ce610H, 30 cm 7.8 mm, Supelco Park, Bellefonte, PA, USA). Organic acids were detected via absorbance at 210 nm, while sugars were detected using

a refractive index detector. Results were expressed as mg 100 mL$^{-1}$ for organic acids and g 100 g$^{-1}$ for sugars of fresh weight (FW). Quantification relied on a standard curve of pure sugars and organic acids obtained from Sigma (Poole, UK). Data were presented as the mean $\pm$ standard error (SE) of three replicates.

### 2.6. Total Phenolic Content (TPC), Hydrophilic and Lipophilic Total Antioxidant Activity

To extract total phenolics, 10 g of berries per cluster were mixed with a solution comprising 30 mL of water:methanol (2:8) and 2 mM NaF. After centrifugation at 10,000$\times$ *g* for 15 min, the resulting supernatant was used for duplicate determinations of total phenolic content employing the Folin–Ciocalteu reagent, following the methodology outlined by Garcia-Pastor et al. [42]. Results were expressed as mg kg$^{-1}$ of gallic acid equivalents on a fresh weight (FW) basis, with data representing the mean $\pm$ standard error (SE) of three replicates.

To evaluate the total antioxidant activity (TAA), 1 g of grape tissue was manually homogenized with 5 mL of 50 mM phosphate buffer at pH = 7.8 and 5 mL of ethyl acetate in a mortar. The resulting mixture was then centrifuged at 10,000$\times$ *g* for 15 min at 4 °C. Subsequently, the upper and lower fractions were used to measure the lipophilic (L-TAA) and hydrophilic (H-TAA) total antioxidant activity, respectively. Duplicate measurements of H-TAA and L-TAA were conducted for each extract using a reaction mixture consisting of 2,20-azino-bis-(3-ethylbenzothiazoline-6-sulfonic acid) diammonium salt (ABTS), horseradish peroxidase enzyme, and hydrogen peroxide as its oxidant substrate. The generation of ABTS+ radicals was monitored at 730 nm. The reduction in absorbance upon addition of the grape extract was directly proportional to the TAA of the sample, quantified using a calibration curve prepared with Trolox (ranging from 0 to 20 nmol) from Sigma Aldrich (Madrid, Spain). The results were expressed as mg of Trolox Equivalent (TE) per 100 g of fresh weight, with the data representing the mean $\pm$ standard error (SE) of three replicates.

### 2.7. Mineral Content

Mineral content (MC) analysis followed the protocol outlined by Lorente-Mento et al. [43]. Duplicate samples, each consisting of 0.25 g of dehydrated berries and leaf tissues obtained from a composite of 10 fruits or leaves per replicate, were prepared. Next, each sample underwent digestion using a microwave digester (CEM Mars One, Matthews, NC, USA) with a 1% HNO$_3$ solution. After digestion, the samples were diluted to a final volume of 50 mL with distilled water. Aliquots were then taken for the determination of macro and microelements using inductively coupled plasma mass spectrometry (ICP-MS) (Shimadzu ICP-MS-2030, Kyoto, Japan). Mineral quantification relied on standard curves for Ca, Cu, Fe, K, Mn, Na, and P. The determination of cell wall-bound Ca$^{2+}$ in berry fruit (three replicates of 20 g tissue) followed the protocol outlined by Michailidis et al. [44]. Fruit tissues were homogenized and boiled in 95% ethanol for 20 min. After centrifugation (11,000$\times$ *g* for 10 min at 4 °C), the supernatant was discarded. The residue underwent multiple centrifugation steps with 80% ethanol (11,000 g for 10 min at 4 °C) until the solution clarified. Then, the residue was washed with pure acetone and centrifuged again (11,000$\times$ *g* for 10 min at 4 °C). Subsequently, the acetone was removed by drying the pellet at 60 °C. The dried pellet was stored until Ca$^{2+}$ analysis, following the previously described method for mineral content analysis. The mineral content was expressed as mg per 100 g of dry weight.

### 2.8. Malondialdehyde (MDA) and Abscisic Acid (ABA) Quantification

Measurement of MDA in grape tissue: Five grams of grape tissue were blended with 5 mL of 10% (*v/v*) trichloroacetic acid (TCA). After homogenization, the mixture underwent centrifugation at 10,000 rpm for 10 min at 4 °C. 2 mL of the resulting extract were combined with 2 mL of 0.67% thiobarbituric acid (TBA) and then heated to boiling at 100 °C for 15 min. The absorbance at 450 nm, 532 nm, and 600 nm was subsequently

measured using a Shimadzu UV-Vis spectrophotometer, model UV-1900, Kyoto, Japan. The concentration of MDA was determined using the following formula:

$$MDA = 6.45 \times (Abs532 - Abs600) - (0.56 \times Abs450). \text{ Results are expressed in } \mu g\, g^{-1}.$$

To assess ABA levels, grapes were initially ground into powder using liquid nitrogen and then mixed with a solution consisting of 80% methanol and 1% acetic acid, containing deuterium-labeled ABA as internal standards. The mixture was vigorously shaken for one hour at 4 °C. Subsequently, the resulting extract underwent overnight incubation at 20 °C, followed by centrifugation. The supernatant was then dried using a vacuum evaporator. The resulting dry residue was dissolved in 1% acetic acid and passed through an Oasis HLB (reverse phase) column (Waters Corp., Milford, MA, USA), following the methodology outlined by Hernández et al. [45]. The dried eluate was reconstituted in 5% acetonitrile–1% acetic acid, and ABA separation was achieved using an auto-sampler coupled with a reverse phase UHPLC chromatographer (2.6 $\mu$m Accucore RP-MS column, 50 mm length $\times$ 2.1 mm i.d.; ThermoFisher Scientific, Waltham, MA, USA). Chromatography involved a gradient of 5–50% acetonitrile containing 0.05% acetic acid, with a flow rate of 400 $\mu$L min$^{-1}$ over 14 min. Hormone analysis was conducted using a Q-Exactive mass spectrometer (Orbitrap detector; ThermoFisher Scientific) through targeted selected ion monitoring (SIM). The concentrations of ABA in the extracts were determined using embedded calibration curves, and data were processed using the Xcalibur 2.2 SP1 build 48 and TraceFinder program version 4.1. Deuterium-labeled hormones were employed as internal standards for ABA quantification.

*2.9. Statistical Analysis*

A two-way ANOVA was performed to evaluate treatment effects, time variations, and their interactions in each experiment. Tukey's test was then employed to detect significant differences among treatments and across different time points. To assess the impact of treatment on grape shattering, mineral content, organic acids, sugars, MDA, and ABA at harvest, a Student's *t*-test was conducted. Statistical significance was determined at $p < 0.05$. The results are presented as means $\pm$ standard error (SE) based on three replicates. Statistical analyses were conducted using IBM SPSS Statistics for Windows version 21.0 (IBM Corp., Armonk, NY, USA).

**3. Results and Discussion**

According to climatological data for the area [39] (Figure 1), 25 days before harvest, in the last days of August and the first fifteen days of September, there was heavy rainfall exceeding 45 mm. This led to a sudden drop in average temperatures from 28 °C to 22 °C and an increase in average relative humidity from 50% to 75%, reaching over 90% at night. During this period, the sky was cloudy with low radiation (less than 20 MJ m$^{-2}$), preventing crop drying. These environmental conditions near harvest time promote berry shattering, cracking, and rot, as noted by Strik [46]. Fourie [47] reported that tropical storms in California during the dry season rapidly increased humidity, leading to insufficient cluster drying due to persistent clouds and warmer temperatures, which contributed to harvest loss.

In this context, 62.26 $\pm$ 2.30% of control vines showed detached berries on the ground at harvest (Figure 2A). However, the percentage of vines with berries detached on the ground was significantly lower ($p < 0.001$) for those treated with sorbitol (34.41 $\pm$ 1.5%), Ca (45.50 $\pm$ 2.01%), or sorbitol–Ca (35.85 $\pm$ 1.20%). In previous years, 2021 and 2022, during August and September, these climatic conditions did not occur; there was no precipitation before harvest, temperatures were more stable, and average relative humidity was below 70%. Consequently, shattering in these vineyards was less severe. Bassiony et al. [48] confirm that calcium applications in the form of Ca reduced cluster shattering at harvest.

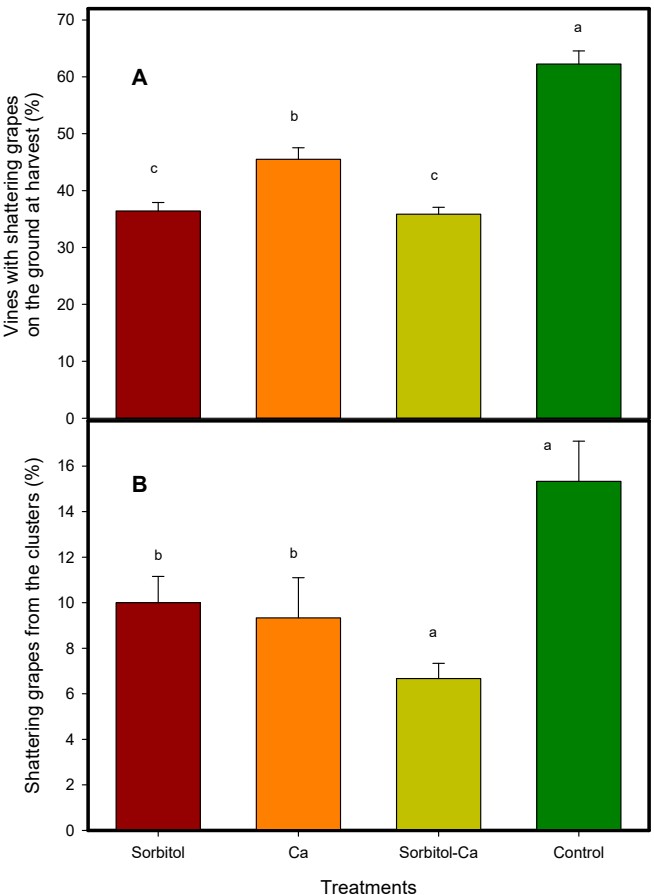

**Figure 2.** Vines with shattering grapes (%) (**A**) and shattering grapes from clusters (%) (**B**) in table grape treated with sorbitol (Red), Ca (Orange), sorbitol–Ca (Yellow), and control (Green) at harvest and at the end of storage. Data are the mean ± SE of three samples for each replicate and treatment. Different letters indicate significant differences between treatments ($p < 0.05$).

On the other hand, berry shattering during storage was consistently higher in control clusters compared to treated clusters (Figure 2B). By the end of storage, there were $15.33 \pm 1.76\%$ detached grapes in the control clusters, while in the treated clusters, it was $10.0 \pm 1.15\%$, $9.33 \pm 1.76\%$, and $6.67 \pm 0.67\%$ for those treated with sorbitol, Ca, and sorbitol–Ca, respectively. Sorbitol–Ca treatment was the most effective in reducing berry detachment. Numerous studies demonstrated the effect of calcium treatments applied as $Ca(NO_3)_2$ on controlling berry drop during postharvest storage, such as Young-Sik et al. [49] in 'Cheongsoo' cultivar and Bassiony et al. [48] in 'Thompson Seedless' cultivar reduced berry drop by 45% when treated with Ca, compared to control grapes. In addition, it is worth noting that sorbitol–Ca treatment was more effective than Ca in reducing berry shattering, which could be related to the higher absorption and mobility of calcium when applied as sorbitol complex than as Ca.

In fact, mineral analysis at harvest revealed that leaves treated from vines with sorbitol–Ca had significantly higher ($p < 0.05$) calcium concentration than leaves from vines treated with sorbitol, Ca, and control (Table 1). However, Ca or sorbitol treatments did not affect calcium concentration in leaves. Additionally, sorbitol–Ca treatment also increased Cu, Fe, Mg, and P concentrations compared to control and Ca-treated leaves. Regarding mineral content in grapes (skin + pulp), sorbitol–Ca treatment also significantly increased ($p < 0.001$) total calcium concentration compared to other treatments and control (Table 2). Additionally, Cu and Fe concentrations in grapes (skin + pulp) treated with sorbitol–Ca and sorbitol were higher ($p > 0.05$) than those treated with Ca and control. In this regard, K/Ca, Mg/Ca, and (K + Mg)/Ca ratios were significantly lower in grapes

treated with sorbitol–Ca compared with other treatments ($p < 0.05$). Furthermore, bound calcium in fruits was significantly higher ($p < 0.001$) in grapes treated with sorbitol–Ca.

**Table 1.** Total minerals in grapevine leaves at harvest. The data are expressed as mg 100 g$^{-1}$ of dry weight and represent the means $\pm$ SE of three replicates. According to Student's *t*-test ($p < 0.05$), different letters between columns (treatments) for the same mineral indicate significant differences.

| Mineral | Control | Ca | Sorbitol | Sorbitol–Ca |
|---|---|---|---|---|
| Ca | 4259.71 $\pm$ 125.03 b | 4364.45 $\pm$ 48.63 b | 4427.73 $\pm$ 88.91 b | 5105.64 $\pm$ 84.11 a |
| Cu | 207.18 $\pm$ 26.89 b | 225.89 $\pm$ 4.51 b | 161.18 $\pm$ 3.00 c | 253.26 $\pm$ 12.59 a |
| Fe | 21.30 $\pm$ 0.49 c | 24.57 $\pm$ 1.93 bc | 26.20 $\pm$ 1.91 b | 36.63 $\pm$ 1.70 a |
| K | 457.05 $\pm$ 13.51 ab | 440.64 $\pm$ 17.41 b | 477.91 $\pm$ 33.49 a | 476.13 $\pm$ 5.84 a |
| Mg | 472.55 $\pm$ 10.77 c | 448.44 $\pm$ 4.72 d | 496.35 $\pm$ 6.87 b | 543.80 $\pm$ 17.60 a |
| Mn | 18.70 $\pm$ 0.38 b | 18.36 $\pm$ 0.36 b | 20.86 $\pm$ 0.24 a | 18.28 $\pm$ 0.43 b |
| Na | 61.56 $\pm$ 2.34 c | 115.44 $\pm$ 6.47 a | 83.49 $\pm$ 5.12 b | 119.73 $\pm$ 5.66 a |
| P | 420.31 $\pm$ 13.16 cb | 410.98 $\pm$ 11.63 c | 439.63 $\pm$ 5.88 b | 491.63 $\pm$ 5.66 a |
| $\Sigma$minerals | 5921.90 $\pm$ 177.27 b | 6052.48 $\pm$ 47.00 b | 6137.16 $\pm$ 49.38 b | 7049.64 $\pm$ 69.73 a |
| Ca + K | 4716.76 $\pm$ 125.28 c | 4805.09 $\pm$ 46.74 c | 4905.65 $\pm$ 55.79 b | 5581.77 $\pm$ 79.42 a |
| K/Ca | 0.11 $\pm$ 0.00 a | 0.10 $\pm$ 0.00 a | 0.11 $\pm$ 0.01 a | 0.09 $\pm$ 0.00 b |
| Mg/Ca | 0.11 $\pm$ 0.00 a | 0.10 $\pm$ 0.00 a | 0.11 $\pm$ 0.00 a | 0.11 $\pm$ 0.00 a |
| (K + Mg)/Ca | 0.22 $\pm$ 0.01 a | 0.20 $\pm$ 0.00 b | 0.22 $\pm$ 0.01 a | 0.20 $\pm$ 0.01 b |
| K/Mg | 0.97 $\pm$ 0.03 a | 0.98 $\pm$ 0.05 a | 0.96 $\pm$ 0.05 a | 0.88 $\pm$ 0.02 b |

**Table 2.** Total minerals and bound calcium in grape fruits at the time of harvest. The data are expressed as mg 100 g$^{-1}$ of dry weight and represent the means $\pm$ SE of three replicates. According to Student's *t*-test ($p < 0.05$), different letters between columns (treatments) for the same mineral indicate significant differences.

| Total Mineral in Grapes | | | | |
|---|---|---|---|---|
| Mineral | Control | Ca | Sorbitol | Sorbitol + Ca |
| Ca | 86.68 $\pm$ 5.89 b | 82.39 $\pm$ 0.94 b | 66.68 $\pm$ 5.60 c | 102.65 $\pm$ 3.67 a |
| Cu | 0.25 $\pm$ 0.01 bc | 0.20 $\pm$ 0.03 c | 0.26 $\pm$ 0.03 ba | 0.31 $\pm$ 0.05 a |
| Fe | 0.68 $\pm$ 0.31 b | 0.61 $\pm$ 0.04 b | 1.53 $\pm$ 0.43 a | 1.09 $\pm$ 0.73 a |
| K | 688.76 $\pm$ 15.37 a | 667.12 $\pm$ 25.35 ab | 645.16 $\pm$ 9.25 b | 638.24 $\pm$ 14.49 b |
| Mg | 39.23 $\pm$ 0.95 a | 34.54 $\pm$ 0.65 b | 33.77 $\pm$ 1.33 b | 35.88 $\pm$ 1.40 b |
| Mn | 0.51 $\pm$ 0.04 a | 0.47 $\pm$ 0.02 a | 0.48 $\pm$ 0.04 a | 0.47 $\pm$ 0.05 a |
| Na | 14.92 $\pm$ 0.30 a | 13.53 $\pm$ 0.38 a | 12.37 $\pm$ 0.30 b | 11.88 $\pm$ 0.81 b |
| P | 218.94 $\pm$ 10.05 a | 211.65 $\pm$ 12.80 a | 195.71 $\pm$ 6.52 b | 191.64 $\pm$ 2.56 b |
| $\Sigma$minerals | 1057.66 $\pm$ 36.46 a | 1010.10 $\pm$ 39.63 a | 955.98 $\pm$ 20.52 b | 983.15 $\pm$ 10.88 ab |
| Ca + K | 775.44 $\pm$ 21.14 a | 749.51 $\pm$ 26.13 a | 711.84 $\pm$ 14.82 b | 740.88 $\pm$ 10.97 a |
| K/Ca | 8.00 $\pm$ 0.39 b | 8.09 $\pm$ 0.24 b | 9.79 $\pm$ 0.66 a | 6.24 $\pm$ 0.35 c |
| Mg/Ca | 0.46 $\pm$ 0.02 ab | 0.42 $\pm$ 0.01 b | 0.51 $\pm$ 0.05 a | 0.35 $\pm$ 0.02 c |
| (K + Mg)/Ca | 8.45 $\pm$ 0.41 b | 8.51 $\pm$ 0.24 b | 10.30 $\pm$ 0.71 a | 6.59 $\pm$ 0.36 c |
| K/Mg | 17.56 $\pm$ 0.04 b | 19.30 $\pm$ 0.38 a | 19.17 $\pm$ 0.90 a | 17.86 $\pm$ 0.95 b |
| Ligated Ca in grapes | | | | |
| Ca | 40.81 $\pm$ 0.67 b | 33.00 $\pm$ 6.32 b | 36.71 $\pm$ 2.68 b | 55.87 $\pm$ 1.65 a |

According to the results, the differences in total calcium and bound calcium concentrations in fruits treated with sorbitol–Ca (43.13 $\pm$ 3.87 mg 100 g$^{-1}$ DW) and control (50.60 $\pm$ 7.88 mg 100 g$^{-1}$ DW) are similar. This indicates that the entirety of the calcium transported to fruits treated with sorbitol–Ca becomes part of the pectins in the cell wall and middle lamella of tissues. Therefore, the sorbitol–Ca treatments applied to the leaves (since the clusters were covered with paper bags) were the only ones that significantly increased total leaf calcium and bound calcium in fruits. Berry abscission is closely linked to the concentration of available calcium. Zhu et al. [50] found that the berry abscission percentage was lower in 'Xiangfei' compared to the variety 'Hutai No. 8' because it had a higher concentration of total and bound calcium. The berry shattering results align with those reported by Bassiony et al. [48], which indicate that foliar application of calcium in grapevines resulted in improved rachis and petiole measurements and a reduction in berry shattering at harvest.

Calcium plays a critical role in table grape fruit quality and physiology. Studies have demonstrated that calcium applications before veraison are effective in controlling decay and maintaining fruit quality during storage [19]. Generally, mineral calcium ($CaCl_2$, $Ca(NO_3)_2$) applied after veraison has limited phloem mobility, forming calcium chelates with minimal arrival at the fruits [19]. However, when calcium is applied after veraison in the form of nanoparticles, with particle sizes less than 100 nm [51], enhanced calcium transport to the fruits occurs, increasing total calcium content and calcium bound to pectins in fruit tissue and reducing berry shattering from the clusters.

Likewise, the application of sorbitol-chelated calcium throughout the crop cycle increased leaf calcium concentration by 13.12–19.32% and kernel calcium concentration by 6.49–8.15% compared to the control in peanuts [52] and total calcium and calcium bound in the leaves and fruits of pepper plants [53]. This chelation of calcium with polyols converts inorganic calcium to organic, enabling a vectoring process for absorption, translocation, and transformation [54]. In this regard, the results demonstrate that calcium applied in the inorganic form of calcium nitrate did not accumulate in leaves or fruit tissues due to its difficulty in being incorporated into leaves and redistributed through the phloem route from leaves to fruit. However, calcium applied in the chelated form significantly increased the free calcium concentration in leaves and fruits, primarily binding to pectins in the cell wall and middle lamella.

The balanced and timely availability of mineral nutrients is crucial for achieving optimal plant performance. In addition to the necessary concentrations of each macro- and micro-element, the proportion between elements also plays a crucial role in growth, productivity, quality, and nutrient uptake. Variations in the K/Ca ratio in berries have been employed as an indicator of alterations in the relative potassium influx through the xylem and phloem [55]. Low ratios of K/Ca, Mg/Ca, and (K + Mg)/Ca in fruits serve as indicators for predicting fruit quality and the occurrence of physiological disorders such as cold damage, sunburn, and fruit cracking, among others [43,56]. In this regard, 'Asgari' grapevines treated with calcium exhibited a lower K/Ca ratio and showed reduced berry drop [57].

The increase in calcium, both total and bound to pectins, allowed for significantly firmer fruit at harvest for sorbitol–Ca treatment ($1.21 \pm 0.03$ N mm$^{-1}$) compared with sorbitol, Ca, and control ($1.11 \pm 0.03$; $1.10 \pm 0.04$; and $0.99 \pm 0.03$ N mm$^{-1}$). Although the firmness of all fruits decreased significantly during storage, those from vines treated with sorbitol–Ca exhibited greater compression resistance (Figure 3A). Preharvest and postharvest calcium treatments in different fruits (climacteric and non-climacteric) have allowed for increased firmness and consequently extended fruit shelf life. However, depending on how calcium is applied or formulated, its effect can vary in efficiency [58,59]. In the present experiment, calcium applied in the chelated form with sorbitol proved to be the most effective.

External calcium supply can promote the formation of non-covalent bonds between pectin molecules by creating calcium bridges. This strengthens the cell wall structure and prevents middle lamella dissolution. As a result, calcium applications often lead to an increase in the fraction of insoluble pectins in the form of pectates, primarily composed of cell wall pectin molecules linked non-covalently, which is associated with better retention of total uronic acids. In addition to its direct effect on cell structure, calcium can enhance firmness by modulating various enzymatic activities (pectinmethylesterase, polygalacturonase) involved in cell wall modification [17,60].

At harvest, all clusters exhibited the same respiration rate. However, the respiration rate and weight loss (Figure 3B,C) of the treated clusters were significantly lower than those of the controls during the storage period. By the end of storage, the control fruits showed significantly higher respiration rate ($19.91 \pm 0.89$ mg $CO_2$ kg$^{-1}$ h$^{-1}$) and weight loss ($4.82 \pm 0.49\%$) compared to the other treatments, with clusters from the sorbitol–Ca-treated vines exhibiting the lowest respiration rate ($15.27 \pm 0.60$ mg $CO_2$ kg$^{-1}$ h$^{-1}$) and weight loss ($3.73 \pm 0.29\%$). Preharvest calcium treatment generally reduces the respiration rate

and, consequently, fruit ripening processes [59]. Similarly, grape varieties 'Perlette' and 'King's Ruby' treated with CaCl$_2$ exhibited a lower respiration and transpiration rate than controls, which was inversely proportional to the applied dose, and treated grapes showed lower weight loss at the end of storage due to enhanced membrane integrity (phospholipid and protein stability) and cell wall stability [61].

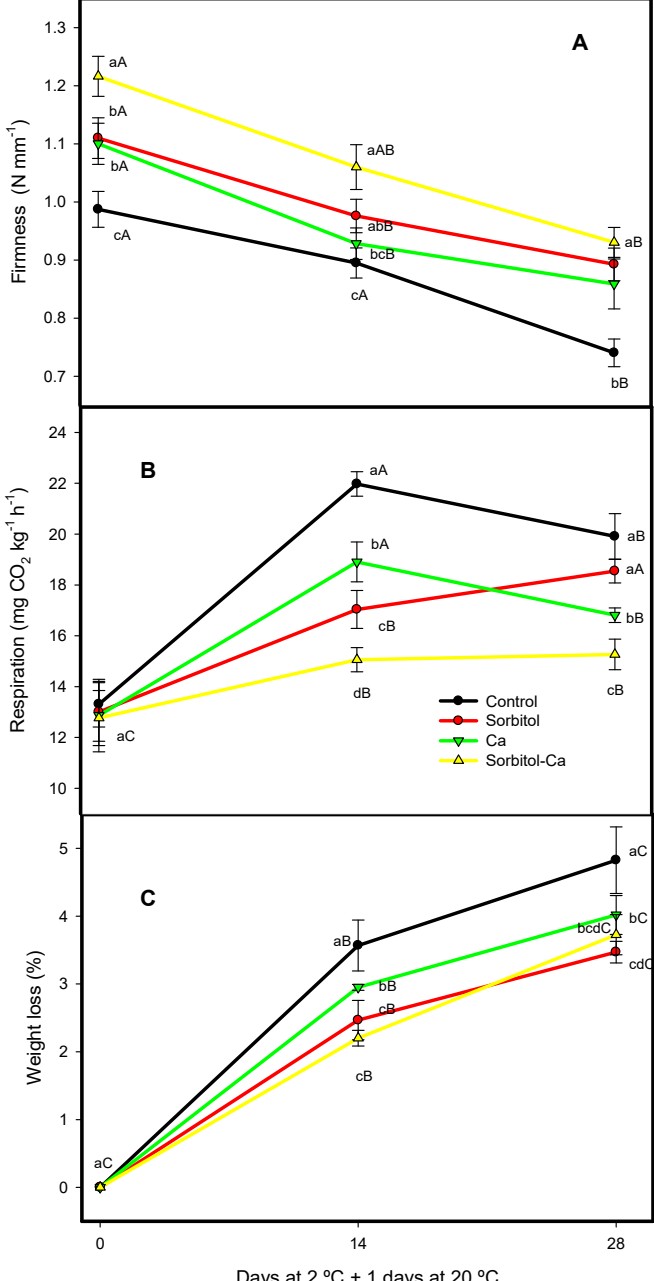

**Figure 3.** Firmness (N mm$^{-1}$) (**A**), respiration rate (mg CO$_2$ kg$^{-1}$ h$^{-1}$) (**B**) and weight loss (%) (**C**) of table grape treated with sorbitol (red), Ca (green), sorbitol–Ca (yellow), and control (black) during storage at 2 °C plus 1 day at 20 °C. Data are the mean ± SE of three samples for each replicate and treatment. Different lowercase letters indicate significant differences between treatments on the same sampling day, and different uppercase letters indicate significant differences between samples of the same treatment on different storage days.

In this context, the MDA (malondialdehyde, which is a major product of lipid peroxidation in membranes) content of the control grapes was significantly higher than that of grapes treated with sorbitol, Ca, and sorbitol–Ca throughout the entire experiment (Figure 4A).

Similarly, grapes of the 'Vinhão' variety [17] and 'Li Xiu' variety [62] treated with calcium exhibited lower MDA concentrations either at harvest or throughout the storage process. In the case of the 'Thompson Seedless' variety treated with calcium nanoparticles, the percentage of detached grapes was directly proportional to the accumulated MDA concentration in grape tissues [51].

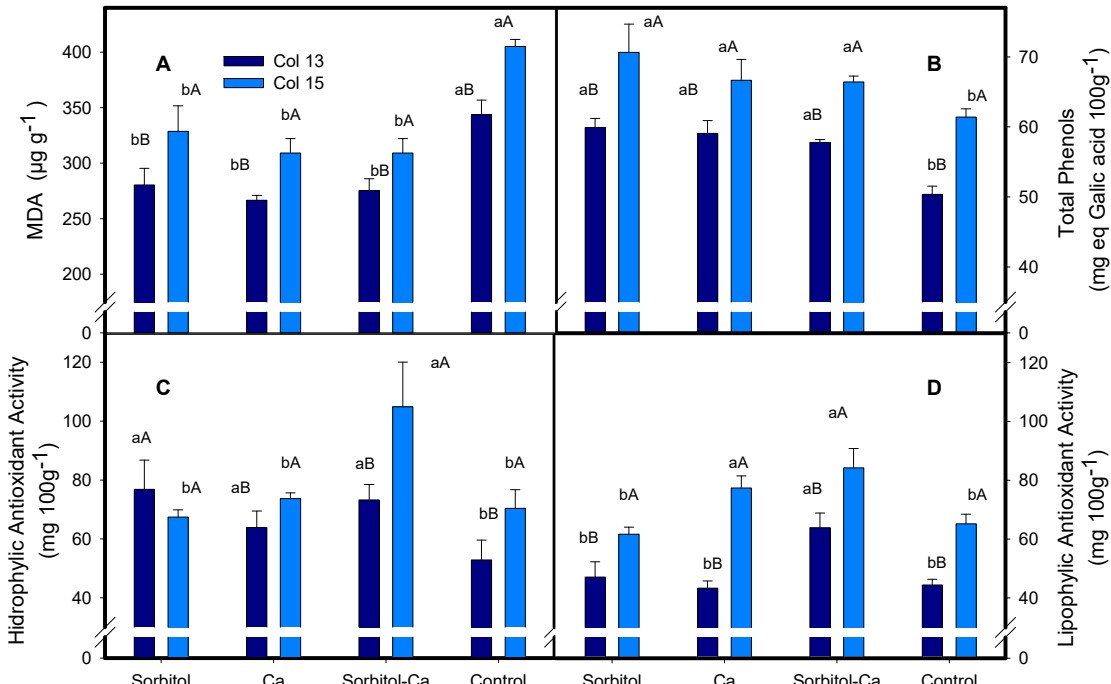

**Figure 4.** MDA ($\mu$g g$^{-1}$) (**A**), total phenols (mg Gallic acid 100g$^{-1}$) (**B**), hydrophilic antioxidant activity (mg 100g$^{-1}$) (**C**) and lipophylic antioxidant activity (mg 100g$^{-1}$) (**D**) in table grape treated with sorbitol, Ca, sorbitol–Ca and control at harvest and the end of storage. Data are the mean $\pm$ SE of three samples for each replicate and treatment. Different lowercase letters indicate significant differences between treatments on the same sampling day, and different uppercase letters indicate significant differences between samples of the same treatment on different storage days.

Nevertheless, unlike MDA, treated grapes exhibited higher total phenol content and hydrophilic and lipophilic antioxidant activity compared to control grapes, both at harvest and throughout storage (Figure 4B–D). Nonetheless, grapes treated with sorbitol–Ca exhibited significantly greater antioxidant activity (both lipophilic and hydrophilic) compared to the other treatments and the control. Similarly, in the white grape variety cv. 'Loureiro', calcium-treated grapes influenced the expression of phenylalanine ammonia lyase and stilbene synthase enzymes, leading to the accumulation of caftaric, coutaric, and fertaric acids, along with specific stilbenoids, such as E-$\omega$-viniferin and E-piceid. Additionally, it improved the concentration of flavonols and flavan-3-ols [63]. Moreover, sorbitol treatments in potatoes increased phenylalanine ammonia-lyase (PAL) activity, as well as suberin and lignin [64]. Sorbitol treatments in maize seedlings conferred greater drought resistance and improved antioxidant systems to mitigate abiotic stress [65]. Conde et al. [66] demonstrated that preharvest treatments with 2 mM sorbitol and 2 mM mannitol increased the concentration of anthocyanins, stilbenes, and total phenolics in grape berries.

On the other hand, sorbitol–Ca treatment significantly reduced ABA accumulation in berries compared to the control at harvest (Figure 5). It is important to note that calcium acts as a secondary messenger with various plant hormones, specifically calcium and ABA, which have multiple interactions in stress responses and fruit ripening regulation [67]. Thus, ABA increases cellulase and polygalacturonase activity, playing a crucial role in berry abscission. However, preharvest treatments with CaCl$_2$ on grapes significantly decreased ABA synthesis and reduced berry drop [9] and cracking [68].

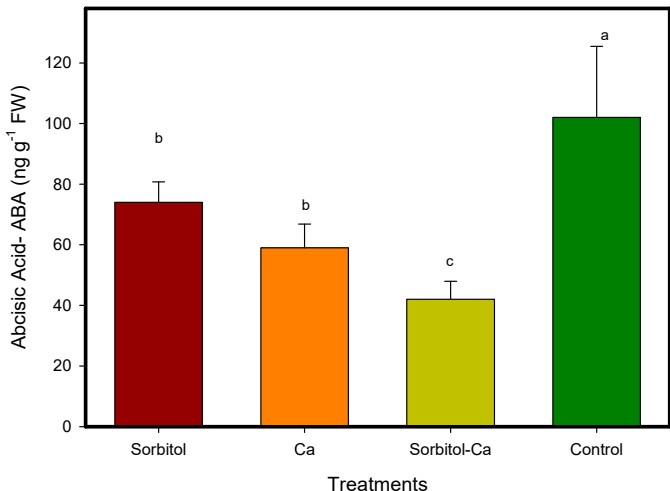

**Figure 5.** ABA content (ng g$^{-1}$) in table grape berries treated with sorbitol, Ca, sorbitol–Ca, and control at harvest. Data are the mean $\pm$ SE of three samples for each replicate and treatment. Different letters indicate significant differences between treatments.

In general, the major organic acids at harvest were tartaric (200–270 mg 100 mL$^{-1}$) and malic in lower concentration (90–160 mg 100 mL$^{-1}$) (Table 3). Citric, fumaric, succinic, and oxalic acids were present in concentrations below 10 mg 100 mL$^{-1}$. Grapes treated with sorbitol, Ca, and sorbitol–Ca showed lower accumulation of tartaric acid, while the concentration of malic acid was significantly higher compared to control fruits, especially in grapes treated with sorbitol–Ca.

**Table 3.** Organic acids and sugars in grapes fruit at harvest. The data are means $\pm$ SE of three replicates. According to Student's *t*-test ($p < 0.05$), different letters between columns (treatments) for the same organic acid or sugar indicate significant differences.

| Organic Acids (mg 100 mL$^{-1}$) | Control | Ca | Sorbitol | Sorbitol + Ca |
|---|---|---|---|---|
| Tartaric | 270.0 $\pm$ 10.0 a | 240.0 $\pm$ 10.0 b | 190.0 $\pm$ 10.0 b | 200.0 $\pm$ 10 c |
| Malic | 90.01 $\pm$ 10.9 c | 160.0 $\pm$ 10.0 a | 140.0 $\pm$ 10.0 b | 170.0 $\pm$ 10.0 a |
| Citric | 20.8 $\pm$ 0.9 a | 18.3 $\pm$ 0.6 b | 17.9 $\pm$ 0.2 b | 21.3 $\pm$ 0.03 a |
| Fumaric | 21.4 $\pm$ 3.1 a | 26.9 $\pm$ 5.6 a | 22.6 $\pm$ 2.7 a | 20.3 $\pm$ 1.1 a |
| Succinic | 14.8 $\pm$ 0.07 c | 16.2 $\pm$ 0.32 a | 15.8 $\pm$ 0.46 abc | 15.6 $\pm$ 0.25 bc |
| Oxalic | 0.15 $\pm$ 0.03 a | 0.13 $\pm$ 0.02 a | 0.11 $\pm$ 0.01 b | 1.03 $\pm$ 0.005 b |
| Total | 420.0 $\pm$ 10.0 b | 460.0 $\pm$ 10.0 b | 370.0 $\pm$ 20.0 a | 430.0 $\pm$ 10.0 b |
| Tart/Malic | 3.24 $\pm$ 0.67 a | 1.49 $\pm$ 0.14 b | 1.42 $\pm$ 0.05 b | 1.18 $\pm$ 0.07 c |
| **Sugars (g 100 mL$^{-1}$)** | **Control** | **Ca** | **Sorbitol** | **Sorbitol + Ca** |
| Glucose | 7.45 $\pm$ 0.34 b | 7.73 0.09 b | 8.25 $\pm$ 0.32 a | 8.12 $\pm$ 0.19 a |
| Fructose | 6.35 $\pm$ 0.26 b | 6.63 $\pm$ 0.11 b | 7.13 $\pm$ 0.33 a | 6.80 $\pm$ 0.11 a |
| Total | 13.80 $\pm$ 0.70 b | 14.36 $\pm$ 0.21 b | 15.38 $\pm$ 0.64 a | 14.92 $\pm$ 0.27 a |
| Gluc/Fruc | 1.18 $\pm$ 0.02 a | 1.17 $\pm$ 0.01 a | 1.16 $\pm$ 0.01 a | 1.19 $\pm$ 0.02 a |

At harvest, the predominant sugars were glucose and fructose, each in concentrations around 0.1 to 0.2 g 100 mL$^{-1}$. Treatments modified the accumulation and synthesis of organic acids and sugars in the fruits (Table 3). Generally, the major organic acids in grapes are malic and tartaric acids [69]. Typically, there is an accumulation of malic and tartaric acids before véraison, after which there is a significant decrease in malic acid content. Tartaric acid, on the other hand, undergoes minimal changes until harvest. Varieties such as 'Red Globe', 'Thompson Seedless', and 'Crimson Seedless' exhibited this behavior during their development. The proportion of organic acids depends on environmental conditions,

and maturity status, among other factors [69], with the effect of sorbitol–Ca treatment primarily responsible for delaying malic acid degradation in our case.

Furthermore, the major sugars in grapes were glucose and fructose, present in proportions close to 54% for glucose and 46% for fructose regardless of the treatment applied (Table 3). However, grapes treated with sorbitol and sorbitol–Ca reached a higher concentration of glucose and fructose compared to control grapes and those treated with Ca.

The total soluble solids (TSS) and maturity index (MI) in fruits treated with sorbitol and sorbitol–Ca were significantly higher than in controls and Ca-treated fruits at harvest and during storage (Figure 6). These results align with those reported by Ma et al. [70], who applied pre-harvest treatments of sugar alcohol-chelated calcium to increase TSS and IM in 'Cabernet Sauvignon' grapes. Generally, pre-harvest treatments with sorbitol alone or sorbitol-chelated calcium improve TSS and MI content in various fruits like mango [71], tomato [36], and wine grape [70]. This effect could be attributed to sugar alcohols facilitating nutrient and reserve transport to plant tissues, especially fruits like grapes, as Conde et al. [72] characterized a sorbitol transporter (VvPLT1) in grape mesocarp cells responsible for sugar alcohol absorption, which could be crucial under stress conditions where sugar alcohols are rapidly oxidized to reducing sugars by sorbitol dehydrogenases. Pre-harvest treatments with polyols (2 mM sorbitol and 2 mM mannitol) on 'Touriga Nacional' grapes doubled sorbitol dehydrogenase activity in mature grapes [66].

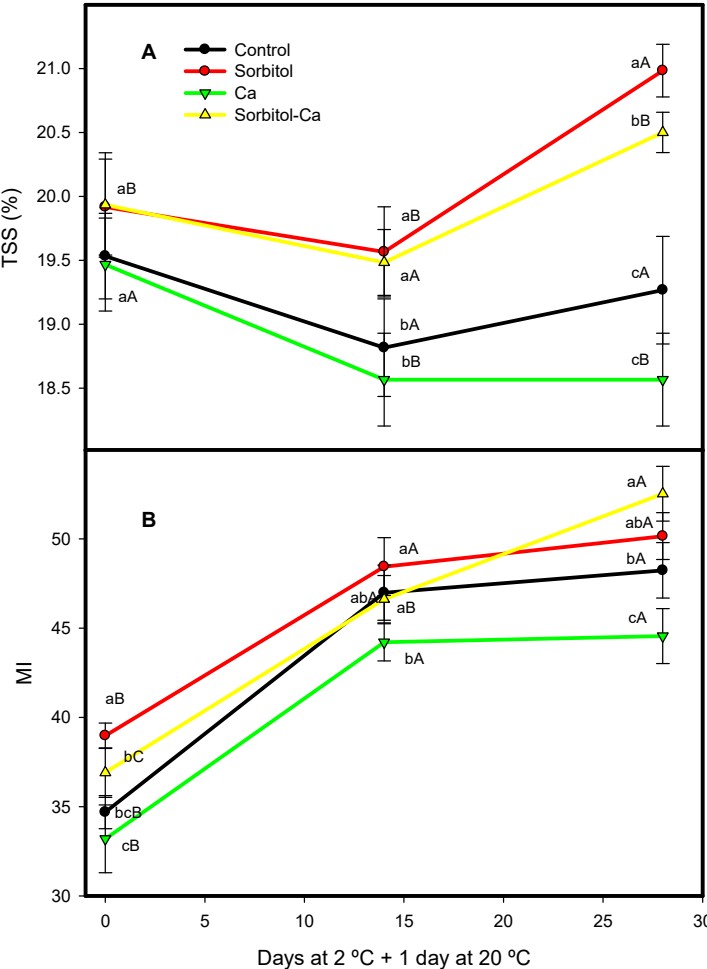

**Figure 6.** TSS (%) (**A**) and maturity index (MI) (**B**) of table grape treated with sorbitol (red), Ca (green), sorbitol–Ca (yellow), and control (black) during storage at 2 °C plus 1 day at 20 °C. Data are the mean $\pm$ SE of three samples for each replicate and treatment. Different lowercase letters indicate significant differences between treatments on the same sampling day, and different uppercase letters indicate significant differences between samples of the same treatment on different storage days.

## 4. Conclusions

This work demonstrates that pre-harvest treatments from veraison to harvest with sorbitol–Ca on bagged table grapes (cv. 'Doña María') reduced berry drop at harvest and during postharvest. This treatment increased calcium transport from the leaves to the berries, while the Ca treatment did not increase calcium concentration in either leaves or fruits. Additionally, part of the calcium accumulated in the fruits bound to the cell wall as calcium pectate, improving their firmness. Furthermore, sorbitol–Ca reduced ripening processes, decreasing fruit quality loss, evidenced by a lower respiration rate, reduced ABA and MDA accumulation, increased total phenolic content, antioxidant activity, glucose and fructose content, as well as reduced malic acid degradation. In this regard, the use of sorbitol–Ca could be considered a safe and efficient tool for grape cultivation to control berry drop in varieties susceptible to this disorder or under abiotic stress conditions, ensuring their successful commercialization.

**Author Contributions:** Conceptualization, D.M.-R., H.M.D.-M. and J.M.V.; methodology, M.S. and D.V.; software, A.G.; validation, M.S., D.V. and D.M.-R.; formal analysis, A.G., D.M.-R., H.M.D.-M. and J.M.V.; investigation, A.G., D.M.-R., H.M.D.-M. and J.M.V.; resources, D.M.-R. and J.M.V.; data curation, D.M.-R.; writing—original draft preparation, A.G., D.M.-R., H.M.D.-M. and J.M.V.; writing—review and editing, M.S., D.V. and D.M.-R.; supervision, D.M.-R., H.M.D.-M. and J.M.V.; project administration, D.M.-R. and J.M.V.; funding acquisition, D.M.-R. and J.M.V. All authors have read and agreed to the published version of the manuscript.

**Funding:** Grant number PID2022-137282OB-I00 funded by MICIU/AEI/10.13039/501100011033 and by FEDER, UE.

**Data Availability Statement:** Data are contained within the article.

**Acknowledgments:** The authors thank Generalitat Valenciana, Conselleria of Education, Universities and Employment for the PhD-scholarships of Alberto Guirao Carrascosa (CIACIF/2022/270) to pursue doctoral studies and to the European Social Fund for co-financing these grants. We also thank the Protected Designation of Origin (PDO) 'Uva de mesa embolsada Vinalopó' for their support and Francisco Javier Barbero Arenas, the owner of the vineyards.

**Conflicts of Interest:** The authors declare no conflicts of interest.

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
