# Peer review of "Role of Pre-Harvest Sorbitol–Calcium Treatments in Controlling Berry Drop in Bagged Table Grapes of the “Doña María” Variety"

_horticulturae, doi:10.3390/horticulturae10070698_

Round 1

Reviewer 1 Report

Comments and Suggestions for Authors

The manuscript by Guirao et al. explored preharvest treatment with Sorbitol-Calcium in controlling fruit drop of table grapes. The manuscript can be accepted after minor revision. Please see my comments to improve the manuscript.

Line 171-172, each vine was sprayed with how much volume of the solutions?

Line 200, what was the temperature of this step?

Line 214, the diameter of the probe was 10 CM? The firmness was measured with peel or not?

Figure 1, I recommend the authors provide the fruit picture of corresponding timepoint.

Line 338-342, please provide the ANOVA analysis information and the meaning of the different order of the labels.

Line 367-369, 370-372, 515-516, please provide the data analysis method and the meaning of different letters in the columns, i.e. who compared with who.

Line 536, please provide the full name of M.I.

Author Response

Response to Reviewer 1 Comments 

Comments and Suggestions for Authors

The manuscript by Guirao et al. explored preharvest treatment with Sorbitol-Calcium in controlling fruit drop of table grapes. The manuscript can be accepted after minor revision. Please see my comments to improve the manuscript.

We express our gratitude to the reviewer for their valuable comments and questions. We have thoroughly revised this manuscript in accordance with the feedback received and the new sentences added have been marked in red ink. The specific points raised by the reviewer are addressed below.

R1.1- Line 171-172, each vine was sprayed with how much volume of the solutions?

It has been indicated on line 172. “Each treatment was carried out in a row of 120 vines with a dose of 1200 L ha-1.”

R1.2-Line 200, what was the temperature of this step?

The sampling of gases inside the containers was conducted at 20°C. This information has been added on line 201.

R1.3-Line 214, the diameter of the probe was 10 CM? The firmness was measured with peel or not?

Yes, a 10 cm probe was used to deform the whole berry (pedicel, flesh, and peel) by 5%. This information has been included in line 215.

R1.4-Figure 1, I recommend the authors provide the fruit picture of corresponding time point.

Thank you very much for your appreciation. In Figure 1, images of the clusters have been included for each treatment at the time of harvesting.

R1.5-Line 338-342, please provide the ANOVA analysis information and the meaning of the different order of the labels.

The legend of figure 2 has been modified

Lines 341-344. “Figure 2. Vines with Shattering grapes (%) and shattering grapes from clusters (%) in table grape treated with sorbitol (Red), Ca (Orange), Sorbitol-Ca (Yelow) and control (Green) at harvest and at the end of storage. Data are the mean ± SE of three samples for each replicate and treatment. Different letters indicate significant differences between treatments (p < 0.05).

R1.6. Line 367-369, 370-372, 515-516, please provide the data analysis method and the meaning of different letters in the columns, i.e. who compared with who.

Thank you very much for your suggestion and appreciation

The titles of Tables 1, 2, and 3 have been modified.

Lines 372-375. “Table 1. Total minerals in grapevine leaves at harvest. The data are expressed as mg 100g-1 of dry weight and represent the means ± SE of three replicates. According to Tukey's test (p < 0.05), minerals with different letters between columns indicate significant differences.”

Lines 369-371.Table 2. Table 2. Total minerals and bound calcium in grape fruits at the time of harvest. The data are expressed as mg 100 g-1 of dry weight and represent the means ± SE of three replicates. According to Tukey's test (p < 0.05), minerals with different letters between columns indicate signifi-cant differences.

Lines 502-504. Table 3. Organic acids and sugars in grapes fruit at harvest. The data are means ± SE of three replicates. According to Tukey's test (p < 0.05), organics acids or sugars with different letters be-tween columns indicate significant differences.

R1.7. Line 536, please provide the full name of M.I

Maturity Index (MI) has been included in the caption of Figure 6.

Reviewer 2 Report

Comments and Suggestions for Authors

The paper entitled Role of Pre-harvest Sorbitol-Calcium Treatments in Controlling Berry Drop in Bagged Table Grapes of the "Doña María" Variety is thematically focused on evaluate whether calcium either applied individually as inorganic calcium (Ca(NO3)2) (Ca) or in combination with sorbitol (Sorbitol-Ca) can translocate to grapes and prevent their detachment from clusters during cultivation and postharvest storage.

The introductory part of the contribution is elaborated very comprehensively and at a standard level, I have a comment about the clear definition of the goals of the experiments (brief and short definition).

The methodological part is processed in a logical sequence with a description of the methods and devices used. Some inaccuracies appear in the results section, for example writing units in index form (line 375, 376, 501). On the other hand, the conclusion is very brief, where the authors could better emphasize the importance of the achieved results. I also recommend performing a language correction. After incorporating these comments, the text can be accepted for publication.

Comments on the Quality of English Language

Minor editing of English language required

Author Response

Response to Reviewer 2 Comments 

Review 2

The paper entitled Role of Pre-harvest Sorbitol-Calcium Treatments in Controlling Berry Drop in Bagged Table Grapes of the "Doña María" Variety is thematically focused on evaluate whether calcium either applied individually as inorganic calcium (Ca(NO3)2) (Ca) or in combination with sorbitol (Sorbitol-Ca) can translocate to grapes and prevent their detachment from clusters during cultivation and postharvest storage.

We express our gratitude to the reviewer for their valuable comments and questions. We have thoroughly revised this manuscript in accordance with the feedback received and the new sentences added have been marked in blue ink. The specific points raised by the reviewer are addressed below.

R2.1.-The introductory part of the contribution is elaborated very comprehensively and at a standard level, I have a comment about the clear definition of the goals of the experiments (brief and short definition).

The methodological part is processed in a logical sequence with a description of the methods and devices used.

Thank you very much for your comments.

R2.2.- Some inaccuracies appear in the results section, for example writing units in index form (line 375, 376, 501).

Thank you very much for your appreciation. The changes have been made. Line 378, 379, 506

R2.3. On the other hand, the conclusion is very brief, where the authors could better emphasize the importance of the achieved results.

The conclusions have been improved. The following text has been added.

Line  539-550. “This work demonstrates that pre-harvest treatments from veraison to harvest with sorbitol-Ca on bagged table grapes (cv. 'Doña María') reduced berry drop at harvest and during post-harvest. This treatment increased calcium transport from the leaves to the berries, while the Ca treatment did not increase calcium concentration in either leaves or fruits. Additionally, part of the calcium accumulated in the fruits bound to the cell wall as calcium pectate, improving their firmness. Furthermore, sorbitol-Ca reduced ripening processes, decreasing fruit quality loss, evidenced by a lower respiration rate, reduced ABA and MDA accumulation, increased total phenolic content, antioxidant activity, glucose and fructose content, as well as reduced malic acid degradation. In this regard, the use of sorbitol-Ca could be considered a safe and efficient tool for use in grape cultivation to control berry drop in varieties susceptible to this disorder or under abiotic stress conditions, ensuring their successful commercialization.”

R3.4.-I also recommend performing a language correction.

Thank you very much.

The language has been reviewed.

Reviewer 3 Report

Comments and Suggestions for Authors

Review of the manuscript entitled „Role of Pre-harvest Sorbitol-Calcium Treatments in Controlling Berry Drop in Bagged Table Grapes of the Doña María Variety”

The reviewed manuscript concerns the possibility of reducing berry drop and improving the quality and durability of table grapes through the pre-harvest application of sorbitol and calcium nitrate. Thanks to chemical analyzes of leaves and fruits, the authors showed that Ca used in a complex with sorbitol is more efficiently taken up by the leaves and transported to the fruit than Ca used only in the mineral form.   Thanks to the analyzes and measurements performed, the authors showed that sorbitol-Ca reduced ripening processes, decreasing fruit quality loss, evidenced by lower respiration rate, reduced ABA and MDA accumulation, increased total phenols, antioxidant activity, glucose and fructose content, as well as reduced malic acid degradation. Experiment design, analyzes and measurements, as well as the method of statistical processing of the results are correct. The introduction contains all the necessary information regarding the topic under study. The methodology and results are well described, and the conclusions are appropriate and justified. The manuscript was well edited and written in correct language. The article has great cognitive and practical value. In my opinion, it may be the subject of further stages of editorial work, in particular after taking into account minor detailed comments.

 Detailed comments

The authors provide the dose of the substances used as a percentage of the spray liquid. Most often, the dose is presented as the weight (volume) of the preparation per 1 hectare of vineyard area or 10,000 m2 LWA (leaf wall area). In the methodology, the authors provide the concentration, volume of spray liquid per vein and the number of veines per 1 ha. Based on this data, the dose of chemicals used per vineyard area or LWA can be calculated. Nevertheless, it is worth providing at least in the abstract, in addition to the concentration, the dose in kg (l)/ha (10,000 m2 LWA).

l. 114-115. Ad. sentence „Fruit calcium content can be classified according to its solubility and physiological activity”. Since calcium is a component of fruit, it means that it was available to plants. I suggest rephrasing this sentence differently, e.g.”Calcium compounds can be classified according to its solubility and physiological activity, including transport possibilities in the plant”.

Date of manuscript received: 3 June 2024

Date of this review: 7 June 2024

Author Response

Response to Reviewer 3 Comments 

We express our gratitude to the reviewer for their valuable comments and questions. We have thoroughly revised this manuscript in accordance with the feedback received and the new sentences added have been marked in green ink. The specific points raised by the reviewer are addressed below.

R3.1.The authors provide the dose of the substances used as a percentage of the spray liquid. Most often, the dose is presented as the weight (volume) of the preparation per 1 hectare of vineyard area or 10,000 m2 LWA (leaf wall area). In the methodology, the authors provide the concentration, volume of spray liquid per vein and the number of veines per 1 ha. Based on this data, the dose of chemicals used per vineyard area or LWA can be calculated. Nevertheless, it is worth providing at least in the abstract, in addition to the concentration, the dose in kg (l)/ha (10,000 m2 LWA).

Thank you very much for your suggestion. You are absolutely right.

The applied dosage of the treatments has been included.

Line 174. “Each treatment was carried out in a row of 120 vines with a dose of 1,200 L ha-1

R3.2. l. 114-115. Ad. sentence „Fruit calcium content can be classified according to its solubility and physiological activity”. Since calcium is a component of fruit, it means that it was available to plants. I suggest rephrasing this sentence differently, e.g.”Calcium compounds can be classified according to its solubility and physiological activity, including transport possibilities in the plant”.

Thank you very much for your accurate remark. This sentence has been added to the text.

Line 114. “Calcium compounds can be classified according to its solubility and physiological activity, including transport possibilities in the plant